# Associations Among Beliefs Supporting Patriarchal Principles, Conflict Avoidance, and Economic Violence in Intimate-Partner Relationships of Ultra-Orthodox Jews

**DOI:** 10.3390/bs14111114

**Published:** 2024-11-20

**Authors:** Ruth Berkowitz, David Mehlhausen-Hassoen, Zeev Winstok

**Affiliations:** 1School of Social Work, Faculty of Social Welfare and Health Sciences, University of Haifa, Haifa 3498838, Israel; 2School of Social Work, Center for Research and Study of the Family, Faculty of Social Welfare and Health Sciences, University of Haifa, Haifa 3498838, Israel; davidmh@univ.haifa.ac.il (D.M.-H.); zwinstok@univ.haifa.ac.il (Z.W.)

**Keywords:** patriarchal principles, conflict avoidance, economic violence, intimate-partner violence

## Abstract

Beliefs that uphold patriarchal principles may influence individuals’ willingness to avoid conflict in their intimate-partner relationships, which can, in turn, increase the likelihood of intimate-partner economic violence. However, these associations remain underexplored in current research. This study examines these dynamics within a sample of 321 adults from the Ultra-Orthodox Jewish community—a patriarchal and traditional culture. Specifically, it examines associations among beliefs supporting patriarchal principles at the micro (gendered domestic roles), meso, and macro (institutional power of men and the inherent inferiority of women) levels; conflict avoidance; economic violence; and sex differences in these factors. Descriptive statistics were used to analyze sex differences in the study variables, and path analysis was used to examine the correlations between research variables for men and women. The findings indicate that beliefs in patriarchal ideologies were moderate across all levels but slightly higher among men. Men were significantly more likely than women to avoid conflict with their intimate partners. Beliefs in support of patriarchal ideologies were predictive of conflict avoidance, particularly among women. Contrary to prior research, this study revealed nonsignificant sex differences in the prevalence of economic-violence victimization. These findings, however, do not negate the role of sex-based dynamics in economic-violence victimization. We discuss the findings and the meanings assigned to conflict avoidance by men and women, while considering gender disparities of power and control. We suggest that men’s tendency to avoid conflict likely moderated their likelihood of perpetrating economic violence.

## 1. Introduction

Beliefs that uphold patriarchal principles play a fundamental role in shaping the behaviors of men and women across all levels of social interaction [1]. These beliefs may differentially affect the behaviors of men and women in their intimate relationships. First, they may affect their willingness to trigger conflict or reluctance to oppose their partner’s position or behavior. Second, such beliefs, coupled with conflict avoidance, may promote or prevent economic violence. This study examined sex differences in beliefs supporting patriarchal principles, conflict avoidance, and economic violence and the associations among these factors. To enable greater understanding of the contribution of support for patriarchal beliefs and issues related to intimate-partner relationships, this study focused on the Ultra-Orthodox Jewish population in Israel. The Ultra-Orthodox Jewish culture is a collectivist, traditional, patriarchal, and insular culture that tends to hold conservative and traditional views of gender roles and intimate-partner relationships [2].

### 1.1. Patriarchal Beliefs

Patriarchy characterizes all human societies to some degree and can be defined as a gendered sexual system in which men exert control over women, and in which what is considered masculine is valued more than what is considered feminine [3]. Research based on feminist approaches has indicated that violence against women is deeply rooted in the patriarchal structures of society [4]. Patriarchy is a multidimensional concept (for further discussion, see [5]). Yoon and colleagues [4] identified three factors representing patriarchal beliefs operating on micro, meso, and macro levels of social systems and integrated them in their Patriarchal Beliefs Scale (PBS) [4]: Macro-level beliefs reinforce male authority through cultural norms and laws. Meso-level beliefs manifest in workplaces, educational institutions, and community leadership. Micro-level beliefs are evident in interpersonal relationships, particularly role expectations within families. This multidimensional approach helps frame the current study’s analysis of patriarchal beliefs in the context of economic violence.

The first factor identified by Yoon et al. [4] is the institutional power of men, referring to beliefs in general male authority and leadership. The underlying assumption of these beliefs is male domination and female subordination, especially at meso and macro levels of social systems and in matters of greater importance and social impact. The second factor is the inherent inferiority of women, referring to beliefs in women’s inferiority, subordinate status, and restriction or exclusion from diverse social roles, mostly at meso and macro levels, like the workplace, education, community involvement, etc. The underlying assumption of these beliefs is male superiority and female inferiority, which should be reflected in their social status and roles. Finally, the third factor is gendered domestic roles, referring to beliefs in gendered roles in the family (micro level), according to which men are destined to be providers and decision-makers and women are destined to be caretakers for children and the household. The underlying assumption of these beliefs is a natural order, an indisputable precondition of women and men for specific familial functions.

#### Theoretical Framework

This study’s examination of patriarchal beliefs is grounded in feminist and social-dominance theories [6]. Feminist theory posits that gender-based violence, including economic abuse, stems from systemic power imbalances that privilege men over women [1,7]. As Dobash and Dobash [7] argued, violence against women is fundamentally rooted in patriarchal social structures. This perspective has been consistently supported by meta-analytic evidence linking patriarchal ideology with intimate-partner violence [1,8].

Each level of patriarchal belief may influence conflict avoidance and economic violence distinctly [4,9]. Macro-level beliefs about male institutional power may legitimize men’s economic control by framing it as natural and appropriate [8,10]. Meso-level beliefs about women’s inherent inferiority may justify employment sabotage and financial exploitation through reduced value placed on women’s economic contributions [11,12]. Finally, micro-level beliefs about gendered domestic roles may directly impact daily financial decision-making and conflict patterns, potentially normalizing economic control within the household [13,14].

### 1.2. Cultural Context of the Study

Although legally and formally an egalitarian, liberal democracy, Israeli society is also characterized by a great variety of ethnicities, religions, and cultures, some of which have more pronounced patriarchal structures and hold more conservative patriarchal beliefs than others [15]. Israeli society features Palestinian Israelis (25.7%) and Jewish Israelis (74.3%). The Jewish Israeli group can be further divided according to religiosity levels, including religious (15.5%), traditional (25%), secular (45.4%), and Ultra-Orthodox (14.1%) populations. This study was conducted among the Ultra-Orthodox population in Israel—a culturally isolated group that has been described in the literature as a collectivist, patriarchal system created and run by men, which is insular and tends to hold conservative and traditional views of gender roles and intimate-partner relations [16]. Ultra-Orthodox Jews have often shown reluctance to acknowledge, or outright resistance to, the idea of feminism, viewing their religious practices as needing to remain constant in the face of secular influences [17]. Ultra-Orthodox women play a significant role as wives and mothers, whereas men are in charge of the public domain and religious practices [18,19]. Despite the seriousness of domestic violence and abuse, there has been little discussion on this topic within the Ultra-Orthodox Jewish community. Ultra-Orthodox individuals, and women in particular, often feel pressured to maintain their community’s image by keeping their families intact, which can force them to endure traumatic situations rather than seek escape. Consequently, they bear the burden of shame and responsibility for their circumstances [20]. Ultra-Orthodox society fosters patriarchal beliefs and structures, which are most pronounced on the micro level of family relations [16]. Because of these unique characteristics, a focus on the Ultra-Orthodox group allows a better understanding of the influence that support for patriarchal beliefs has upon intimate-partner relationships, including conflict avoidance and economic violence.

### 1.3. Partner Conflict Avoidance

Willingness to engage in and avoidance of conflict in intimate relationships have been studied in recent decades [21]. A distinction is frequently made between two conflict styles: demand (which involves actively engaging in conflict through criticism and complaints) and withdrawal (which entails avoiding conflict through defensiveness and passivity). Prior research has identified three distinct dyadic patterns [22]. Two of these patterns are symmetrical, where both partners exhibit the same style (i.e., demand–demand or withdrawal–withdrawal), while the third pattern is asymmetrical (i.e., demand–withdrawal). The asymmetrical pattern has been noted for its destructive potential [23], leading to greater scrutiny in studies. Evidence suggests that, within an asymmetrical pattern, women are more likely to adopt a demanding approach, while men tend to withdraw [24].

The interaction between patriarchal beliefs and conflict avoidance can be understood through both feminist theory [5,7] and social-dominance theory [25]. Feminist theory suggests that women’s tendency toward conflict engagement may represent resistance against patriarchal control, whereas men’s withdrawal maintains power without obvious displays of dominance [13,26]. Social-dominance theory would frame this conflict-avoidance pattern as hierarchy-maintaining behaviors that preserve existing power structures [27]. These patterns reflect broader systemic power dynamics [1], in which conflict avoidance behaviors serve to maintain hierarchical structures within intimate relationships [7,21]. As Sagrestano and colleagues [13] demonstrated, such interaction patterns are intrinsically linked to perceived power differences between partners.

Various perspectives have been proposed to explain sex differences in this context: the individual-differences perspective [28], the gender-role and socialization-differences perspective [29], the power-discrepancies perspective [26], and the issues-and-goals perspective [30]. According to the latter perspective, it is primarily a partner’s position and goals regarding a specific issue that determine whether an individual will demand or withdraw from a conflict regarding that issue. Put simply, when spouses desire changes in their relationship or partner, they are more likely to criticize and demand. In contrast, when spouses are satisfied with the status quo, they are more likely to avoid the discussion and withdraw [22]. Accordingly, due to the sex power balance and men’s tendency toward preserving rather than changing this balance, men are stipulated to display a weaker inclination than women to initiate intimate conflicts, whereas women’s tendency to initiate such conflicts is stronger. Moreover, men’s tendency to evade partner conflicts could be rewarding by generating a false belief that the man is not the dominant party in the relationship [21]. Additionally, subject to the issues-and-goals perspective [30] and the sex power balance, in the existing social constructs, men arguably have a clear advantage over women outside the family home. This advantage makes the home a woman’s last stronghold. Therefore, they will strive more than men to protect their rights therein, even via conflict, if required.

### 1.4. Economic Violence

In May 2011, economic violence was added as a fourth form of violence against women—after physical, sexual, and psychological violence—in Article 3 of the Council of Europe’s [31] Istanbul Convention. Nevertheless, the concept of economic violence remains under development. Through a feminist theoretical lens [5,7], economic violence represents a manifestation of patriarchal control at multiple levels: macro (i.e., systemic economic inequalities), meso (i.e., workplace discrimination and access to community resources), and micro (i.e., household financial control). These multilevel manifestations of control [4,8] align with established patterns of patriarchal power dynamics, in which economic domination serves as a mechanism for maintaining gender-based hierarchies within intimate relationships [13,14]. Such economic control tactics operate systematically across social domains [10,11], ranging from institutional barriers to individual financial decision-making. A recent meta-analysis [8] demonstrated that economic abuse has not always been clearly defined and that its measurement varied substantially across 46 studies. However, some characteristics of economic violence have been rather consistently represented across many studies.

One guiding theoretical framework is the Scale of Economic Abuse [10], which identifies two distinct dimensions of economic abuse: economic control and economic exploitation. Economic control occurs when the husband achieves control over the family’s finances, for example, by taking over his wife’s bank account, limiting her access to money or other means of payment, or demanding a full report of her spending. It also includes tactics that interfere with the wife’s ability to acquire resources or use her existing resources. Economic exploitation, on the other hand, occurs when the husband coerces his wife into financial obligations, such as by acquiring debt or spending money in her name, or leaving her responsible for bills vital to running the household or child-rearing. It also includes tactics of monitoring the wife’s usage of resources and dictating how she uses them, or depleting her of resources altogether.

Another conceptual framework [11], based on the experiences of victims of economic abuse, identifies three common abusive tactics utilized by men against their wives, adding employment sabotage as the third tactic of economic violence. Employment sabotage occurs when the perpetrator inhibits his wife from acquiring training or education, seeking employment, or succeeding in her job. This enhanced conceptual framework was empirically tested and led to the development of the shortened, three-factor version of the Scale of Economic Abuse utilized in the current study [11].

Economic partner violence harms women financially and disrupts their efforts to be economically independent [12]. Various aggressive behaviors are intended to “control a woman’s ability to acquire, use and maintain economic resources, thus threatening her economic security and potential for self-sufficiency” [10] (p. 564). Economic exploitation, economic control, and employment sabotage have been identified as the most noticeable tactics of economic violence [8,10]. This research assesses economic violence in terms of these tactics.

### 1.5. Summary and Research Hypotheses

The study tested the following hypotheses: (1) Beliefs supporting patriarchal principles will be upheld by men more than by women. The smallest differences between men and women will be found on the micro level of the practiced role division at home, because major perceived differences between partners in the domestic context may interfere with the functioning of the couple and family. (2) Based on findings of previous studies of partner-conflict avoidance [21], conflict avoidance will be more prevalent among men than among women. (3) Stronger beliefs supporting patriarchal principles will be associated with increased economic violence against women. Further, conflict avoidance by one partner may be considered by the other partner as permission to use power and establish control. Hence, conflict avoidance by one partner will predict the other partner’s use of violence, including economic violence.

## 2. Materials and Methods

### 2.1. Study Procedure and Research Ethics

Data were collected by Ultra-Orthodox university students, who received training for this task from the principal investigators of the study. These students recruited adult contacts, explaining the study’s objectives and emphasizing its significance. Those who expressed their willingness to contribute were then asked to complete questionnaires (either printed or electronic, based on the participant’s preference) privately and anonymously, ensuring that no one would know that they had participated in the study or how they responded. Participants were also informed of their right to decline participation in the study, their option to not answer some questions, and their ability to withdraw at any stage. This study was approved by the institutional review board of the authors’ affiliated institution.

### 2.2. Measurements

Questionnaires included items measuring beliefs in patriarchal ideologies, conflict avoidance, economic violence victimization, and sociodemographics.

#### 2.2.1. Belief in Patriarchal Ideologies

Measurement of belief in patriarchal ideologies was based on a shortened version of the PBS [4]. The original instrument includes 35 items that measure the respondent’s degree of agreement with patriarchal ideologies situated at three levels of social systems: micro level (family, domestic roles; e.g., “Cleaning is mostly a woman’s job”), meso level (school, local community; e.g., “Women should be paid less than a man for doing the same job”), and macro level (state politics, major companies; e.g., “I am more comfortable with men running big corporations than women”). Responses were given on a 5-point Likert scale: 1 = *strongly disagree*, 2 = *slightly disagree*, 3 = *agree to some extent*, 4 = *agree*, and 5 = *strongly agree*. Since the current study focused on additional topics, we used a shortened version of the PBS that included 22 items, to increase the likelihood that participants would complete the questionnaire. Reliabilities of the shortened PBS instrument, as indicated by internal consistency, were high for patriarchal ideologies situated at the micro (α = 0.90), meso (α = 0.71), and macro (α = 0.93) levels. Three summary measures of chronicity were computed, indicating the average of the items measuring patriarchal ideologies situated at the micro, meso, and macro levels.

#### 2.2.2. Partners’ Micro-Level Beliefs Supporting Patriarchal Ideologies

The questionnaire also included items that assessed participants’ perceptions of their partners’ micro-level beliefs regarding patriarchal ideologies. The 10 items that measured participants’ micro-level self-evaluations of patriarchal ideologies were used for this measure, with wording modified so that they would refer to the respondents’ respective partners. A summary measure of the chronicity of partner beliefs was then computed, indicating the average of the 10 items measuring partners’ patriarchal ideologies at the micro level.

#### 2.2.3. Couple Micro-Level Climate

Based on the two micro-level beliefs in patriarchal ideologies (self-evaluations and partner evaluations), a couple-climate variable was computed. This variable represents the average of respondents’ self- and partner evaluations of micro-level beliefs in patriarchal ideologies (α = 0.94).

#### 2.2.4. Conflict Avoidance

A short conflict-avoidance measure consisted of four items. Participants were asked to report the frequency with which they took the following actions when they disagreed with their partner: (a) “You refrained from expressing your opinion, so as not to start a fight with your partner”; (b) “You agreed with your partner’s position, even if you thought it was wrong, to please him/her”; (c) “You accommodated your partner’s demands, so as to not make him/her angry”; and (d) “You did what your partner wanted although it was against your best judgement”. Response options were on a 5-point Likert scale: 1 = *never happens*, 2 = *rarely happens*, 3 = *does not happen a lot, but also not too rarely*, 4 = *often happens*, and 5 = *happens all the time*. Following the collection of data, reliability (internal consistency) was tested for the four items together and found to be high (α = 0.92). A summary measure was computed, indicating the average of the four items measuring conflict avoidance.

#### 2.2.5. Economic-Violence Victimization

Economic-violence victimization was measured utilizing the 12 items of the Revised Economic Violence Scale [11]. The instrument was adjusted for the Hebrew-speaking population and rephrased to accommodate both men and women. Two items were added to the original instrument, with the aim of including essential issues excluded from the original scale (“Your partner made you go to work” and “Your partner neglects household economic obligations”). Hence, the instrument used in this study consisted of 14 items representing various types of economic violence against one’s partner. Each item inquired about the frequency of a specified behavior over the past year (e.g., “Your partner demanded to see what you spent money on” or “Your partner withheld information concerning your finances”). Responses were made on a 5-point Likert scale: 1 = *never happened*, 2 = *rarely happens*, 3 = *does not happen a lot, but also not too little*, 4 = *often happens*, and 5 = *happens all the time*. Reliability for the 14 items was high (α = 0.82).

The second stem tested the factorial construct of the measurement (principal component analysis for extraction and varimax rotation with Kaiser normalization; eigenvalues greater than 1 were deemed significant). The analysis was first performed separately for men and women, then for both sexes together. The analyses were consistent, yielding four factors: (a) monitoring and control (e.g., “Your partner demanded to know what you spent money on”), (b) nondisclosure (e.g., “Your partner made financial decisions without consulting you first”), (c) work-related interference and abuse (e.g., “Your partner prevented you from going to work”), and (d) exploitation and humiliation (e.g., “Your partner wasted the money you needed to use to pay the household bills”). Reliabilities were high for monitoring and control (α = 0.80), nondisclosure (α = 0.78), and work-related interference and abuse (α = 0.79), and fair for exploitation and humiliation (α = 0.65). Based on the measurement analyses, five summary measures were computed, indicating the average of the items that loaded on each factor. The fifth variable was based on the average value of all 14 items (α = 0.82).

#### 2.2.6. Demographics

Participants were requested to provide information regarding their sex (male or female), age, marital status (never married, married, divorced, or widowed), number of children, and educational attainment (master’s degree or higher, bachelor’s degree, 14 years of schooling, high school diploma, or elementary school), economic status (higher than average, average, or lower than average), the extent to which their household income is sufficient for their current expenses (insufficient, partly sufficient, sufficient, and very sufficient), and their degree of household economic stress (high, medium, or low). Since data were collected from the Ultra-Orthodox Jewish population in Israel (by students who are members of this group), there was no need to collect data on participants’ nationality or religiosity.

### 2.3. Analytic Procedure

The first step of the analysis examined the inference and descriptive statistics to identify any sex differences in the research variables. The second step examined the research model, using path analysis to assess correlations between research variables for men and for women. The analyses were conducted using SPSS (version 27) and AMOS (version 27) statistical software.

## 3. Results

### 3.1. Sample Characteristics

This sample included 321 participants, 60.0% of whom identified as female, with an average age of 34.36 years (*SD* = 10.15). Participants reported that they had been married for 11.87 years on average (*SD* = 10.32) and had about three children on average (*SD* = 2.8). Most women (88.3%) and men (91.7%) had higher-education degrees. More than half of the women and men reported that they worked full time (57.8% and 58.4%, respectively) or part time (26.7% and 12.3%, respectively). About half of the participants reported an average economic status (57.4%), that their household income was sufficient for their current expenses (49.5%), and that they experienced medium (43.0%) or high (49.4%) levels of economic stress.

### 3.2. Sex Differences in Beliefs in Patriarchal Ideologies

Sex differences in beliefs in patriarchal ideologies were tested using a repeated-measures procedure. The mean of beliefs in patriarchal ideologies served as the dependent variable. The ecological level at which beliefs were situated (micro, meso, or macro) served as a within-subject variable and respondents’ sex (male or female) served as a between-subject variable. The main effect of the ecological level of beliefs in patriarchal ideologies [*F*(2, 316) = 330.02, *p* < 0.001, η^2^ = 0.68] and the respondents’ sex [*F*(1, 317) = 12.21, *p* < 0.001, η^2^ = 0.04] were significant. The interaction between the ecological level and respondents’ sex was also significant [*F*(2, 316) = 3.78, *p* < 0.024, η^2^ = 0.02]. These findings demonstrate a significant difference between men and women in beliefs in patriarchal ideologies.

To further examine sex differences in beliefs in patriarchal ideologies at each ecological level, *t*-tests were employed. The results are presented in Table 1. Overall, beliefs in patriarchal ideologies were low. The lowest level of beliefs in patriarchal ideologies was observed at the meso level. At the macro and meso levels, men had significantly stronger beliefs in patriarchal ideologies than women. No significant sex differences were found at the micro level.

#### 3.2.1. Sex Differences in Micro-Level Patriarchal Ideologies and Couple Climate in Support of Patriarchal Ideologies

The respondents also provided their estimation of their partner’s support of patriarchal ideologies at the micro level (sex roles). A repeated-measures procedure was used to compare men’s and women’s self-evaluations and partner evaluations of beliefs in patriarchal ideologies. The findings indicate a significant interaction effect between the respondent’s sex (between-subject variable: male or female) and the object of evaluation (within-subject variable: self or partner) for the micro-level beliefs supporting patriarchal ideologies [*F*(1, 313) = 9.69, *p* < 0.002, η^2^ = 0.03]. These results are presented in Figure 1. All participants, but particularly women, estimated their partner’s micro-level beliefs supporting patriarchal ideologies to be higher than their own. Mean comparisons among men’s self-reports of beliefs in patriarchal ideologies and women’s estimations regarding their male partners, and vice versa, revealed similar disparities. Paired-sample *t*-tests showed that the mean difference between self-evaluation and partner evaluation was significant only for women [*t*(190) = 5.77, *p* < 0.001].

Next, sex differences in reports of the couple climate were examined. The *t*-test results are presented at the bottom of Table 1. This analysis revealed that the observed sex differences were not significant.

#### 3.2.2. Correlations Between Couple Climate and Beliefs in Support of Patriarchal Ideologies

The correlations between the couple climate and subscales of beliefs in support of patriarchal ideologies are presented in Table 2. Correlations were strong and significant. Therefore, the couple-climate variable was included in further analyses.

### 3.3. Conflict Avoidance

Sex differences were tested at the average level based on conflict avoidance. Paired-sample *t*-tests revealed significant sex differences in the average level of conflict avoidance, *t*(216.17) = 5.8, *p* < 0.001, demonstrating significantly greater conflict avoidance among men (*M* = 2.42, *SD* = 0.96) than among women (*M* = 1.82, *SD* = 0.78).

### 3.4. Economic Violence Victimization

Sex differences in economic violence victimization were tested based on the four victimization variables (i.e., monitoring and control, nondisclosure, work-related interference and abuse, and exploitation and humiliation) using independent-sample *t*-tests (Table 3). The findings reveal nonsignificant sex differences in economic violence victimization.

### 3.5. Correlations Between Subscales of Economic Violence Victimization

The correlations between subscales of economic violence victimization and economic violence were tested further. These analyses revealed positive and significant correlations (see Table 4).

### 3.6. Associations Between Study Variables

The final study model tested the associations between the study variables using path analysis. The model included three factors: (a) the micro-level couple climate in support of patriarchal ideologies, (b) conflict avoidance, and (c) the four economic violence victimization subscales. To further explicate the study model, multigroup analysis was employed to test whether the hypothesized patterns of interrelationships remained consistent or differed across subsamples by sex. The study model and results are presented in Figure 2. The model’s fit indexes were good [χ^2^(7) = 10.75, *p* = 0.15, NFI = 0.97, IFI = 0.99, CFI = 0.99, RMSEA = 0.04].

The path between couple climate in support of micro-level patriarchal ideologies and conflict avoidance was positive and significant for both men and women, although stronger for women. The stronger the couple’s beliefs in support of patriarchal ideologies, the greater their tendency to avoid conflict. Almost all paths between conflict avoidance and the different subscales of economic violence victimization were significant and positive for both men and women, except for the path between avoidance and nondisclosure among men.

The model findings further indicate sex differences in the paths between couple climate in support of micro-level patriarchal ideologies and the different types of economic violence victimization. For women, the couple climate had a significant positive effect on monitoring: The stronger the couple’s beliefs in support of patriarchal ideologies, the greater the monitoring victimization. For men, the couple climate had significant positive effect on exploitation and humiliation: The stronger the couple’s beliefs in support of patriarchal ideologies, the greater the exploitation and humiliation victimization.

## 4. Discussion

Beliefs that uphold patriarchal principles may influence behaviors in intimate relationships, particularly regarding individuals’ willingness to avoid conflict, which is a factor that can ultimately increase the likelihood of intimate-partner economic violence. However, these associations remain insufficiently explored. To address this gap, this study examined sex differences in beliefs supporting patriarchal principles, conflict avoidance, and economic violence, as well as the associations among these factors, within a sample of Ultra-Orthodox Jewish adults. The Ultra-Orthodox culture has been characterized in the literature as a patriarchal system, in which women are relegated to traditional domestic roles while men dominate the public sphere and religious practices [32]. Results based on scientific exploration of this patriarchal culture can further our understanding of the contribution of beliefs supporting patriarchal ideologies to partner conflict avoidance and economic violence, a relatively new topic of research, and guide interventions to address this concerning social issue.

### 4.1. Beliefs in Support of Patriarchal Ideologies

Overall, participants reported modest levels of belief in patriarchal ideologies at the micro, meso, and macro levels, indicating that individuals in the current sample tended to reject and renounce patriarchal ideologies. The findings further suggest that the extent to which individuals renounce patriarchal ideologies is related to sex: Comparison of beliefs in patriarchal ideologies by sex demonstrated that men’s tolerance and support for patriarchal principles was significantly greater than women’s. It is likely that men expressed greater endorsement of patriarchal ideologies because those ideologies offer men more power and privilege in our society than women [9]. Renouncing patriarchal ideologies may be interpreted by men as a threat to their privilege and dominance.

The findings further suggest that for both men and women, beliefs in patriarchal ideologies were highest at the macro and micro levels and lowest at the meso level. Macro-level patriarchal beliefs reflect societal attitudes regarding the institutional power of men and beliefs in male authority and leadership that go beyond domestic, interpersonal, or work situations [33]. Meso-level patriarchal beliefs regard how such beliefs in male authority and leadership should be applied at the social level (e.g., in the workplace). The participants in the current study tended to agree with the idea of sex differences in authority and power, but at the same time, they were reluctant to apply those differences. In the domestic arena, sex differences were supported through a more traditional division of labor. It seems that the division of traditional domestic roles is perceived as a contract or understanding between spouses that is unrelated to sex differences in authority or power. The findings may also suggest a significant gap between the participants’ rhetoric and practice: Although participants tended to spurn patriarchal ideologies at the meso level, they also tended to support and perhaps implement these patriarchal practices at the micro level in their domestic and interpersonal relationships.

### 4.2. Conflict Avoidance

Advocates of male-control theory [7,34] have argued that in patriarchal social structures, men tend to use aggression and control in intimate relationships to a much greater extent than women. Yet the current findings suggest a different pattern: Men, significantly more than women, tended to avoid conflict with their female intimate partners. These findings echo prior research indicating a stronger escalatory tendency in intimate-partner conflicts among women, as compared to men [21]. A possible explanation for the current finding is that there are sex differences in conflict avoidance. In adherence with the authority and power imbalance by which men enjoy greater control and influence women [13], it can be assumed that men perceive conflict avoidance as fortitude and strength, whereas women perceive it as a weakness. It may be that the privilege of power and authority experienced by men in their intimate-partner relationships encourages their responsibility to maintain their relationships and avoid conflict, if needed. Put differently, when men avoid conflict with their intimate female partner, they display themselves as sensitive to their partner’s needs and desires. Further distinctions can be made to develop a more nuanced understanding of men’s motivations to avoid conflicts and aggression toward women within traditional patriarchal cultures. In patriarchal cultures undergoing transition and change, such as the Arab Muslim culture in Israel, where men often maintain conservative attitudes while women adopt more liberal viewpoints, there is a greater likelihood of disagreements and conflicts arising between spouses [35]. Conversely, in a patriarchal culture where both men and women adhere to traditional gender roles—such as in Ultra-Orthodox Jewish culture—conflicts and disagreements among spouses are less likely to occur. Furthermore, Ultra-Orthodox men are expected to refrain from reacting violently, even when provoked [36]. The Ultra-Orthodox community places a strong emphasis on values such as peace, humility, and self-control. Traditionally, these beliefs discourage violent responses, with non-violence viewed as a virtue indicative of religious devotion. In addition, many religious teachings within Judaism advocate for peaceful conflict resolution and discourage violence, particularly when it comes to maintaining harmonious relationships within the domestic sphere [17,36]. The Ultra-Orthodox conception of masculinity, along with cultural beliefs and community expectations, imposes a mandate on Ultra-Orthodox men to avoid violent behavior. This may lead them to assert power in intimate relationships through economic means rather than through direct verbal or physical confrontation.

It is important to mention that although home and intimate-partner relationships are important for men and women alike, men have more opportunities to fully implement their power in the workplace and public sphere and are less restricted to the domestic arena. As opposed to men, conflict avoidance for women may be perceived as a concession that likely predicts further concessions and undermines their status in their major stronghold—their home. Thus, women invest much of their efforts to safeguard their domestic status and power in their intimate relationships and, accordingly, they do not tend to concede or compromise regarding conflict with their intimate male partner. As a result, sex differences in conflict avoidance create and preserve a false consciousness of sex equity: Men feel highly egalitarian for avoiding conflict with their female intimate partner and women feel highly egalitarian for subordinating or forcing their desires on their male intimate partner.

Consistent with these results, the current findings further suggest that beliefs sustaining patriarchal principles contribute to conflict avoidance for both sexes, albeit to a greater extent among women. Stronger beliefs in patriarchal principles predicted greater conflict avoidance among both men and women—although, again, support of patriarchal ideologies empowers men and inhibits women [37]. Thus, empowered men (who support patriarchal ideologies) do not need to exert strength and demonstrate their dominance in their intimate relationships to establish their status and, therefore, are likely to avoid conflict. On the other hand, men who feel disempowered (who do not support patriarchal ideologies) may need to enhance their domestic status through strength exertion. Within the collectivist, patriarchal, and traditional Ultra-Orthodox culture, disempowered women (who also support patriarchal ideologies) do not recognize their right to exert their strength and so avoid conflict. On the other hand, empowered women (who do not support patriarchal ideologies), like empowered men (who support patriarchal ideologies), do not need to exert strength. Therefore, sex empowerment through men’s support of or women’s rejection of patriarchal ideologies reinforces intimate-partner conflict avoidance.

### 4.3. Economic Violence Victimization

Contrary to prior research [14], the current findings did not indicate any significant sex differences in economic violence victimization. These findings, however, do not suggest that economic violence victimization is unrelated to sex-based dynamics. There may be different explanations for the nonsignificant sex differences in economic violence victimization obtained in the current research. First, economic violence likely disrupts the everyday life of both the perpetrator and victim, causing economic damage. Second, in egalitarian and Western societies, intimate-partner violence, including economic violence, is viewed with contempt and may result in legal restrictions that reduce the perpetrators’ freedom and social status. Yet individuals in the traditional and patriarchal Ultra-Orthodox society will likely hide their victimization, in order to maintain their community’s positive image, rather than flee their traumatic circumstances [20]. It has been suggested that it is only through more nuanced and culturally sensitive understanding by law enforcement and domestic-violence organizations of the uniqueness of this patriarchal conservative culture that domestic violence in the Jewish Orthodox community will be eradicated [38]. The third, and perhaps strongest factor, that moderates men’s and women’s likelihood of perpetrating economic violence against their intimate partner is the positive association between willingness to escalate conflict and perpetration of economic violence. As mentioned, men tend to avoid conflict significantly more than women. Conflict avoidance moderates men’s tendency to perpetrate violence of any kind, including economic violence. Women tend not to avoid conflict and, as a result, become more vulnerable to their intimate partner’s economic violence. Within Ultra-Orthodox society, various mechanisms operate to silence victims of domestic violence. These include passive bystanders who refrain from speaking out about the violence they witness, pressure to maintain the community’s positive image, and resistance to the idea of feminism along with a significant lack of domestic violence-related services [17]. These factors constitute substantial barriers within Orthodox Jewish and other traditional communities that must be addressed more thoroughly. Battin [17] suggests that the recent widespread outcry regarding abusers within the Ultra-Orthodox community, spurred by feminist efforts following the #MeToo movement, may indicate a gradually shifting cultural atmosphere. However, activists worldwide are keenly aware that a larger struggle lies ahead for efforts to achieve lasting and meaningful change.

The insignificant sex differences in economic violence victimization observed in the current research might also be related to the underreporting of victimization by males. Sociocultural norms surrounding masculinity promote ideals of strength, self-reliance, and stoicism in men, which can create stigma and feelings of shame associated with being perceived as a victim [39,40]. As a consequence, these norms can hinder men from reporting their own victimization. It may also be that economic violence manifests in a gender-specific manner. Nevertheless, cultural norms and expectations regarding masculinity and femininity may perpetuate the idea that economic responsibilities and hardships are gender-neutral issues. For instance, societal norms may dictate that men are inherently supposed to provide for the household financially, while women are expected to manage household budgets, thereby obscuring the nuances of how economic violence can vary based on gender. Similarly, cultural stigma around being a victim of violence, in general, and economic violence, in particular, can lead individuals to hide their experiences, especially if those experiences do not conform to traditional gender roles [41].

### 4.4. Study Limitations and Directions for Future Research

While informative, the current findings should be interpreted with certain limitations in mind. First, the study focused on a nonrepresentative sample of Ultra-Orthodox Jewish adults, which means that the findings may not accurately reflect the beliefs and behaviours prevalent within the broader adult population. Although the concentrated examination of the conservative and traditional Ultra-Orthodox culture provided valuable insights into the role of patriarchal ideologies in shaping intimate-partner relationships, it does not shed light on how these dynamics operate in more egalitarian and liberal contexts. To address this limitation, future research should include comparative studies that examine the associations among beliefs supporting patriarchal principles, intimate-partner conflict, and economic violence across diverse cultural settings. Representative samples encompassing a wide range of cultural backgrounds are essential for generalizing findings to the entire adult population. Understanding how these associations manifest in more egalitarian and liberal cultures, as opposed to traditional and patriarchal ones, could reveal critical ethnocultural disparities and variations in relationship dynamics. By exploring these associations, researchers may be able to identify whether patriarchal beliefs, conflict avoidance, and economic violence function differently across cultural spectrums, as well as specific vulnerabilities and strengths inherent to each group. This knowledge is particularly important for creating effective prevention and intervention strategies tailored to the distinct needs of various populations. Ultimately, conducting comparative research will not only expand the understanding of intimate-partner violence within the context of patriarchal principles, but also inform the adaptation of existing interventions to better suit specific cultural frameworks. This approach is essential for enhancing the effectiveness of strategies aimed at reducing intimate-partner violence and promoting healthier relationship dynamics across diverse communities.

Second, the current study’s data indicate associations between variables rather than causal relationships, as the data were collected at a single point in time. Longitudinal data are necessary to substantiate causal claims based on large population-based samples. Thus, future research is encouraged to utilize longitudinal study designs to investigate causal relationships informed by the relevant models and theories.

Future studies should examine the effects of various background variables, such as age, economic status, and educational level, as well as additional family factors, including relationship duration and number of children. This approach could enhance our understanding of how these background variables impact the research model used to investigate the associations among beliefs supporting patriarchal principles, conflict avoidance, and economic violence in intimate-partner relationships.

Moreover, the research model analyzed here is just one of several that could be explored to deepen our understanding of the mechanisms linking the variables addressed in the present study. It is particularly recommended that future research consider models in which certain variables act as moderators or mediators. For instance, testing a model in which conflict avoidance mediates the relationship between patriarchal beliefs and economic status would be valuable.

The current findings revealed nonsignificant sex differences in economic violence victimization. These results align with previous evidence suggesting that men experience partner abuse at rates comparable to women [42,43]. However, research on men’s victimization by their intimate partners remains limited. Therefore, future studies should further explore the lived experiences of men who have faced intimate-partner violence, particularly economic victimization, to provide insight into their perceptions and emotions, as well as the effects of cultural and internalized stigma regarding their experiences. A qualitative approach could be particularly beneficial for gaining a deeper and perhaps more reliable understanding of these issues and potentially shifting cultural norms regarding men’s experiences with intimate-partner and economic victimization. Additionally, cross-cultural research is essential to investigate how various cultural and societal norms influence men’s experiences and perceptions of intimate-partner and economic victimization. Such studies could explore variations across different communities, focusing on how cultural beliefs about masculinity and victimhood impact reporting and coping mechanisms, and they could also examine the roles of patriarchy and intimate-partner dynamics across cultures.

### 4.5. Clinical Implications

The findings have important implications for practice. Social workers, therapists, and other professionals working with couples addressing intimate-relationship violence should recognize that both men and women may exhibit violence in their intimate relationships, not just women. Contrary to the prevailing stereotype that women are more likely than men to be victims of intimate-partner violence, and in contrast to the argument that men in patriarchal social cultures tend to use aggression in intimate relationships to a significantly greater extent than women [6,7], the findings suggest that both partners are vulnerable to economic violence. The Gender Motivation Theory may offer an alternative explanation for the observed gender symmetry in intimate partner victimization [21]. This framework posits that varying social expectations for men and women drive their aggressive behaviors toward intimate partners: Women are driven by a desire to mitigate risk, while men are motivated by the pursuit of status [44]. In situations perceived as unsafe, the pressure for men to enhance their status dictates that they demonstrate resilience, courage, and a readiness to engage in conflict. Conversely, in non-risky situations, such as disagreements with an intimate partner, the same pressure encourages men to exhibit restraint, sensitivity, and caution, fostering a willingness to yield and withdraw [21]. Therefore, interventions should target economic violence regardless of the victims’ sex. Further, to increase the effectiveness of interventions, professionals are encouraged to consider conflict avoidance in intimate relationships as an important predictor of violence victimization.

### 4.6. Policy Implications

Several policy implications and intervention strategies may also emerge from the current findings. First, it is essential to develop and implement educational initiatives aimed at both men and women that challenge patriarchal ideologies and promote greater gender equality. These programs should emphasize the importance of fostering healthy, equitable relationships along with open communication, conflict resolution, and mutual respect between partners. There is also a need to enhance access to support services for victims of economic violence, ensuring these services are sensitive to gender-related issues while addressing the specific needs of all individuals affected. Developing such services is particularly important within the Ultra-Orthodox Jewish community, in which there is a significant deficiency of domestic violence-related resources [17]. Providing culturally sensitive training for social workers, counselors, spiritual leaders, and other professionals within the Ultra-Orthodox community is essential for effectively recognizing the signs of economic violence and understanding how patriarchal beliefs may influence both conflict avoidance and victimization.

## Figures and Tables

**Figure 1 behavsci-14-01114-f001:**
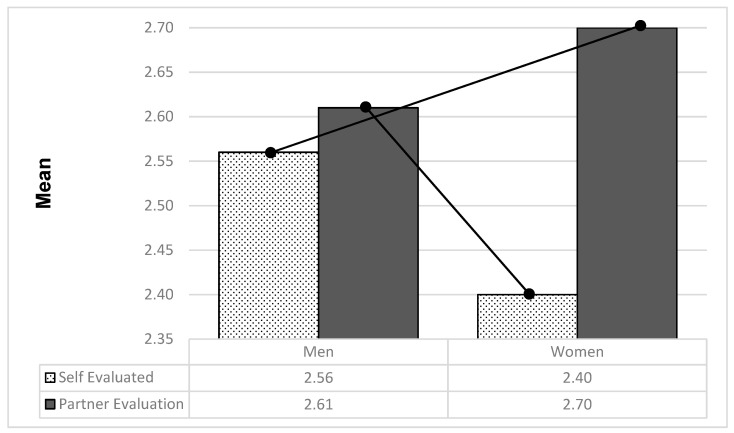
Effect of interaction between respondent sex (male or female) and object of evaluation (self or partner) on micro-level beliefs in support of patriarchal ideologies (traditional sex roles).

**Figure 2 behavsci-14-01114-f002:**
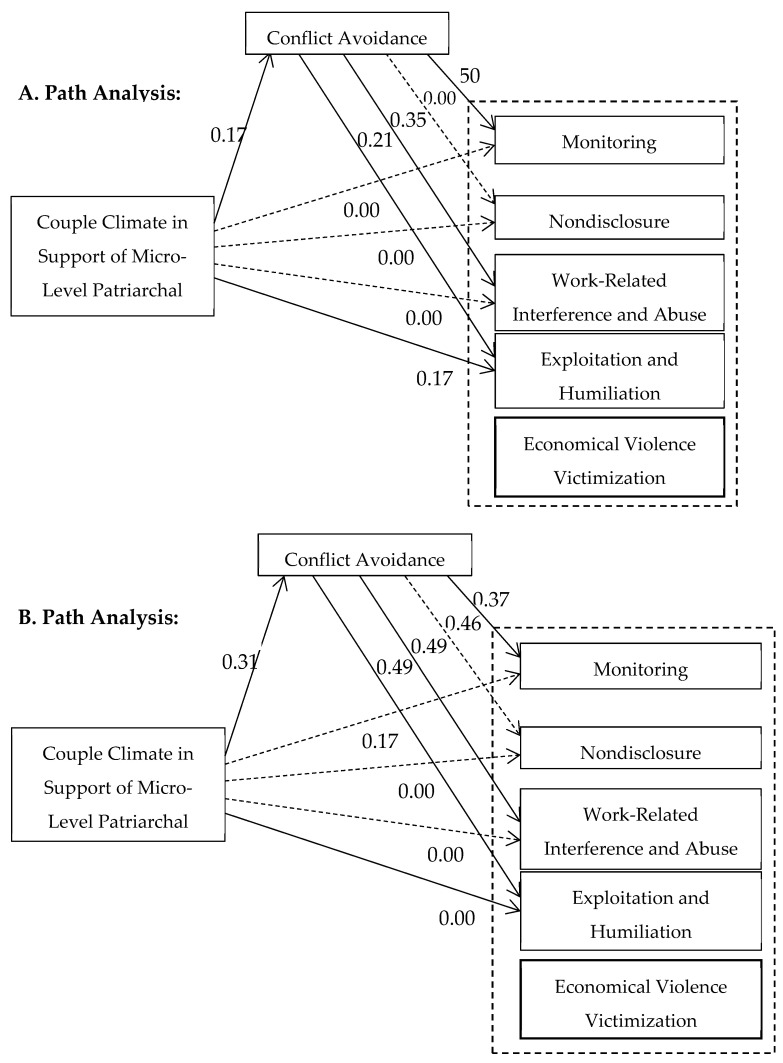
Structural-equation modeling multigroup path analysis.

**Table 1 behavsci-14-01114-t001:** Sex differences in mean levels of beliefs in patriarchal ideologies at the macro, meso, and micro levels.

Variable	*M* (*SD*)	*t* (*df*)	*p*
*Macro*		3.53 (317)	<0.001
Men	2.69 (1.03)		
Women	2.27 (1.06)		
*Meso*		4.81 (205.7)	<0.001
Men	1.59 (0.62)		
Women	1.28 (0.43)		
*Micro*		1.62 (317)	>0.05
Men	2.57 (1.06)		
Women	2.39 (0.95)		
*Partner micro*		0.70 (313)	>0.05
Men	2.61 (1.06)		
Women	2.70 (1.06)		
*Couple climate micro*		0.54 (313)	>0.05
Men	2.60 (1.00)		
Women	2.54 (0.95)		

**Table 2 behavsci-14-01114-t002:** Correlations between the subscales of supporting beliefs of patriarchal principles.

Belief Level	1	2	3	4
1. Macro				
2. Meso	0.70 *			
3. Micro	0.73 *	0.61 *		
4. Partner micro	0.55 *	0.40 *	0.77 *	
5. Couple climate micro	0.67 *	0.53 *	0.94 *	0.94 *

* *p* < 0.001.

**Table 3 behavsci-14-01114-t003:** Sex differences in chronicity of mean economic violence victimization.

Subscale	*M* (*SD*)	*t* (*df*)	*p*
*Monitoring*		0.59 (310)	>0.05
Men	1.61 (0.80)		
Women	1.56 (0.86)		
*Nondisclosure*		1.96 (307.69)	>0.05
Men	1.29 (0.53)		
Women	1.44 (0.78)		
*Work-related interference and abuse*		1.27 (310)	>0.05
Men	1.15 (0.34)		
Women	1.10 (0.37)		
*Exploitation and humiliation*		0.30 (310)	>0.05
Men	1.25 (0.41)		
Women	1.26 (0.46)		
*All subscales*		0.01 (310)	>0.05
Men	1.27 (0.33)		
Women	1.27 (0.37)		

**Table 4 behavsci-14-01114-t004:** Correlations between the subscales of economic violence victimization.

Belief Level	1	2	3	4
1. Monitoring				
2. Nondisclosure	0.31 *			
3. Work-related interference and abuse	0.36 *	0.37 *		
4. Exploitation and humiliation	0.30 *	0.48 *	0.39 *	
5. All subscales	0.66 *	0.71 *	0.73 *	0.79 *

* *p* < 0.001.

## Data Availability

The data is available on request from the authors.

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
