# Peer review of "Associations Among Beliefs Supporting Patriarchal Principles, Conflict Avoidance, and Economic Violence in Intimate-Partner Relationships of Ultra-Orthodox Jews"

_behavsci, 2024, doi:10.3390/bs14111114_

Round 1
Reviewer 1 Report
Comments and Suggestions for Authors
Thank you for the opportunity to study and review this amazing piece of writing.
This is an original and extremely interesting topic, which will certainly excite and attract those with an academic and research interest in this topic.
The structure, discourse, methodology, analysis and presentation of results are all very well structured and presented with clarity and clarity.
There is logical and academic continuity and consistency between the modules, while the length of each module follows a specific logic. In particular, the results are presented both descriptively and schematically with clarity and correctness.
The use of the English language is very good, and the bibliographic references are sufficient.
My only remark concerns the enrichment of the bibliography with more up-to-date and modern bibliographic references.
Author Response
Reviewer 1 comment that the manuscript is focused on an original and engaging topic that would attract individuals with academic and research interests. They noted that the structure, discourse, methodology, analysis, and presentation of results are well-organized and clearly articulated. The reviewer highlighted the logical continuity between sections and the coherent length of each module, with results presented both descriptively and schematically. They commended the quality of the English language used and found the bibliographic references adequate.
The reviewer suggested enriching the bibliography with more current references.
Response: We would like to express our gratitude to the reviewer for their thorough understanding of the importance of this research. To adhere with the reviewer’s suggestion, we updated references and citations throughout the manuscript such that we now use more recent references. These changes are highlighted in yellow in the references list. We did, however, include important older references, such as those by Dobash and Dobash, as they represent classic sources in the literature.
Reviewer 2 Report
Comments and Suggestions for Authors
This is an interesting, informative and generally well-done study. I have two observations: first, I think use of the term "violence" is somewhat misleading. The study does not deal with physical or sexual violence, perhaps for understandable reasons. In the case of psychological violence, this is potentially present in the sense of one partner exercising a degree of control over the other. But what is called economic violence here, does not seem to me to be violence at all, but rather a kind of control. Perhaps the authors would rethink what is meant by violence? Secondly, I would just reinforce the suggestion that doing a similar comparative study with a more liberal adult population could be enlightening in a number of respects.
Author Response
Reviewer 2 mentions that is an interesting, informative and generally well-done study. They suggested that the use of the term "violence" is somewhat misleading. They assert that the study does not deal with physical or sexual violence, perhaps for understandable reasons. In the case of psychological violence, this is potentially present in the sense of one partner exercising a degree of control over the other. But what is called economic violence here, does not seem to me to be violence at all, but rather a kind of control. Perhaps the authors would rethink what is meant by violence?
Secondly, the reviewer reinforces the suggestion that doing a similar comparative study with a more liberal adult population could be enlightening in a number of respects.
Response: We thank reviewer 2 for the thorough reading and constructive suggestion. Indeed, the concept of economic violence is yet developing. Through a feminist theoretical lens, economic violence represents a manifestation of patriarchal control at the micro, meso and macro levels. Our research adheres with this concept. As indicated in our literature review, a relevant guiding theoretical framework is the Scale of Economic Abuse (Postmus et al., 2016[1]). According to this view, economic abuse features two distinct dimensions: economic control and economic exploitation. Thus, economic violence or abuse constitutes both control and exploitation. In adherence with prior research referring to economic abuse as a type of violence (e.g., Postmus et al., 2020[2]), we believe that referring to economic abuse in our study reflects economic violence.
As to suggestions for comparative research in more liberal population, we included a suggestion to conduct similar study in such a population, emphasizing the additional contributions that such comparative research could provide. This addition appears in the Study Limitations and Directions for Future Research section (p. 13, Line 543).
[1] Postmus, J. L., Plummer, S. B., & Stylianou, A. M. (2016). Measuring economic abuse in the lives of survivors: Revising the Scale of Economic Abuse. Violence against women, 22(6), 692-703.
[2] Postmus, J. L., Hoge, G. L., Breckenridge, J., Sharp-Jeffs, N., & Chung, D. (2020). Economic abuse as an invisible form of domestic violence: A multicountry review. Trauma, Violence, & Abuse, 21(2), 261-283.
Reviewer 3 Report
Comments and Suggestions for Authors
Thank you very much for the opportunity to read this important article.
I have a few minor comments and suggestions for improvement and strengthening of the research. I don’t expect the researcher to implement all of them, but I would be pleased to see how they might integrate justifications into their study—explaining why these elements were not included in the research.
To enhance this research on the relationship between patriarchal beliefs, conflict avoidance, and economic violence within the ultra-Orthodox Jewish community, consider the following suggestions:
- Clarify the Conceptual Framework: Explicitly define each level of patriarchal belief (micro, meso, and macro) in the introduction and explain how each might impact conflict avoidance and economic violence. Including theoretical frameworks like feminist theory or social dominance theory could strengthen the interpretation of findings related to power dynamics.
2. I would expect a more in-depth discussion that situates the phenomenon within the Jewish discourse on sexuality and gender. After all, any religious response in the Orthodox community is not isolated from trends and phenomena occurring in secular and non-Orthodox religious communities. I would be pleased to see how this research positions itself in relation to other studies that have addressed religious responses to gendered or sexual expressions. See for instance:
Ben-Lulu, E. (2021). “Teach Your Daughters to Wail and One Another to Lament”: Jewish Prayers and Liturgical Texts for Female Victims of Sexual Assault. Open Theology, 7(1), 631-653.
Kravel-Tovi, M. (2020). “They must join us, there is no other way”: Haredi activism, the battle against sexual violence, and the reworking of rabbinic accountability. Nashim: A Journal of Jewish Women's Studies & Gender Issues, (37), 66-86.
Battin, R. S. (2022). Feminism and# MeToo in the Lives of Orthodox Jewish Women.
- Include Comparative Analysis: To provide context, consider comparing findings with other patriarchal or religious communities. This comparison could highlight unique or common patterns in patriarchal beliefs and their association with conflict avoidance and economic violence across cultures.
- Examine Role of Socialization: Explore MORE how gender socialization within ultra-Orthodox communities shapes attitudes toward conflict and economic control. This could include an analysis of educational practices, religious teachings, and family dynamics.
- Analyze Intersectionality: The role of age, socioeconomic status, and education level could be considered to explore whether these factors interact with gender and patriarchal beliefs in predicting conflict avoidance and economic violence. Intersectional analysis could uncover how diverse experiences within the ultra-Orthodox community affect these behaviors.
- Deepen Statistical Analysis: Expand beyond descriptive statistics and path analysis by considering moderation and mediation analysis to examine if and how conflict avoidance mediates the relationship between patriarchal beliefs and economic violence, with sex as a moderator. This approach could strengthen the understanding of underlying mechanisms.
- Refine Discussion on Sex Differences: Since the study found nonsignificant sex differences in economic violence victimization, discuss these findings in light of past research. Suggest alternative explanations, such as underreporting by men or cultural factors that conceal economic violence as non-gendered, and consider proposing future research to clarify these results.
- Contextualize Conflict Avoidance: Add more context on why conflict avoidance might be higher among men in the ultra-Orthodox community. For example, consider whether religious or cultural teachings on family roles and harmony influence men’s decisions to avoid conflict, potentially leading them to express power in economic ways rather than direct confrontation. See:
Hakak, Y. (2016). Haredi masculinities between the Yeshiva, the army, work and politics: The sage, the warrior and the entrepreneur (Vol. 27). Brill.
- Expand Literature Review: Ensure the literature review covers research on economic violence in similar cultural contexts and integrates cross-cultural studies on patriarchy and intimate partner dynamics. This can help contextualize the findings and address potential limitations in generalizability.
- Policy and Intervention Recommendations: Based on the findings, consider adding a section on policy implications or intervention strategies. You actually did in page#12 – but it's not enough. This could focus on culturally sensitive approaches to reduce economic violence and support conflict resolution within ultra-Orthodox communities.
I believe that each of these suggestions\comments can help to create a more comprehensive, culturally nuanced, and analytically rigorous study.
Author Response
We are grateful for the reviewer’s critical and insightful comments and suggestions, which have significantly contributed to the improvement of the manuscript. The reviewer asserts that they are grateful for the opportunity to read this important article. They suggest that there are a few minor comments and suggestions for improvement and strengthening of the research. The reviewer indicates that they do not expect the researcher to implement all of these suggestions, but they would be pleased to see how the researcher might integrate justifications into their study—explaining why certain elements were not included in the research. They suggest that to enhance the research on the relationship between patriarchal beliefs, conflict avoidance, and economic violence within the Ultra-Orthodox Jewish community, the researcher should consider the following suggestions:
Reviewer 3: Explicitly define each level of patriarchal belief (micro, meso, and macro) in the introduction and explain how each might impact conflict avoidance and economic violence. Include theoretical frameworks like feminist theory or social dominance theory could strengthen the interpretation of findings related to power dynamics.
Response: We defined micro, meso and macro levels of patriarchal believes in the introduction, explained how each might affect conflict avoidance and economic violence, using the frameworks of the feminist and the social dominance theories. These contents appear in the Theoretical Framework (p. 2, Line 72), the Partner Conflict Avoidance (p. 3, Line 122) and Economic Violence sub-sections (p. 4, Line 154).
Reviewer 3: I would expect a more in-depth discussion that situates the phenomenon within the Jewish discourse on sexuality and gender. After all, any religious response in the Orthodox community is not isolated from trends and phenomena occurring in secular and non-Orthodox religious communities. I would be pleased to see how this research positions itself in relation to other studies that have addressed religious responses to gendered or sexual expressions. Explore more how gender socialization within Ultra-Orthodox communities shapes attitudes toward conflict and economic control. This could include an analysis of educational practices, religious teachings, and family dynamics.
Response: We have broadened our discussion on how gender, gender socialization and sexuality within the Ultra-Orthodox Jewish culture is related to domestic violence, conflict and patriarchal believes. We also elaborated on the religious responses to gendered or sexual expressions (p.3, Line 100; p. 12, Line 503; p. 13, Line 515). We used most of the references suggested by the reviewer as they were very helpful in elaborating this interesting discussion.
Reviewer 3: To provide context, consider comparing findings with other patriarchal or religious communities. This comparison could highlight unique or common patterns in patriarchal beliefs and their association with conflict avoidance and economic violence across cultures.
Response: To the best of our knowledge, this study is the first to investigate patriarchal beliefs, conflict avoidance, and intimate partner economic violence within a patriarchal or religious context. We believe that this focus makes our research both unique and significant to the scientific literature. If there are additional studies of which we are unaware, we will be pleased to include them in the manuscript.
Reviewer 3: The role of age, socioeconomic status, and education level could be considered to explore whether these factors interact with gender and patriarchal beliefs in predicting conflict avoidance and economic violence. Intersectional analysis could uncover how diverse experiences within the Ultra-Orthodox community affect these behaviors.
Response: The purpose of the study was to explore associations among patriarchal principles, conflict, and economic violence in intimate partner relationships. We agree with the reviewer that numerous factors influence these associations and merit further examination. Consequently, this necessitates the inclusion of a theoretical explanation in the manuscript's research background, along with a series of additional analyses. However, the intersectionality analyses proposed by the reviewer are beyond the scope of this study. Nevertheless, in light of the importance of this comment, we have added recommendations for future research that include examining how background variables such as age, economic status, and educational level impact the research variables of the current study (p. 14, Line 572).
Reviewer 3: Expand beyond descriptive statistics and path analysis by considering moderation and mediation analysis to examine if and how conflict avoidance mediates the relationship between patriarchal beliefs and economic violence, with sex as a moderator. This approach could strengthen the understanding of underlying mechanisms.
Response: We acknowledge this valuable suggestion. However, as outlined in our response to the previous comment, its inclusion in the manuscript would necessitate a broader theoretical framework to justify the proposed models and additional analyses, which would exceed the manuscript's page limit. Nonetheless, we have incorporated this suggestion into our recommendations for future research as well (p. 14: Line 578).
Reviewer 3: Since the study found nonsignificant sex differences in economic violence victimization, discuss these findings in light of past research. Suggest alternative explanations, such as underreporting by men or cultural factors that conceal economic violence as non-gendered, and consider proposing future research to clarify these results.
Response: Thank you for these critical and relevant suggestions to enhance our discussion surrounding gender issues and intimate partner violence / economic violence. Accordingly, we have elaborated the discussion on additional possible explanations for the insignificant sex differences in economic victimization that emerged in the study, based on underreporting of victimization by males and cultural norms that conceal victimization, and economic violence victimization in particular (p. 13: Line 526). We have also included directions for future research based on these explanations (p. 14, Line 584).
Reviewer 3: Add more context on why conflict avoidance might be higher among men in the Ultra-Orthodox community. For example, consider whether religious or cultural teachings on family roles and harmony influence men’s decisions to avoid conflict, potentially leading them to express power in economic ways rather than direct confrontation. See:
Hakak, Y. (2016). Haredi masculinities between the Yeshiva, the army, work and politics: The sage, the warrior and the entrepreneur (Vol. 27). Brill.
Response: We have elaborated on Ultra-Orthodox conception of masculinity, cultural beliefs and community expectations, which imposes a mandate on Ultra-Orthodox men to avoid violent behavior. We highlight how consequently, this may lead them to assert power in intimate relationships through economic means rather than through direct verbal or physical confrontation (pp. 11-12: Line 546).
Reviewer 3: Ensure the literature review covers research on economic violence in similar cultural contexts and integrates cross-cultural studies on patriarchy and intimate partner dynamics. This can help contextualize the findings and address potential limitations in generalizability.
Response: With the addition of references throughout the literature review, it now more thoroughly addresses and describes the Ultra-Orthodox culture in contrast to modern egalitarian culture, exploring how these cultural differences influence patterns of patriarchal beliefs, conflicts, intimate partner violence, and economic violence specifically. We have also included a discussion on the need for further cross-cultural research. (p. 14: Line 593).
Reviewer 3: Based on the findings, consider adding a section on policy implications or intervention strategies. You actually did in page#12 – but it's not enough. This could focus on culturally sensitive approaches to reduce economic violence and support conflict resolution within Ultra-Orthodox communities.
Response: We have added a section regarding policy implications based on the current findings (p. 15, Line 610).
Reviewer 4 Report
Comments and Suggestions for Authors
Line 47-49: There seems to be a need to define this concept very well for the flow of discussion;
Section 1.3 Partner Conflict Avoidance : This discussion depicts what is happening in many other societies where some avoid unnecessary stress as well as learning more democratic gender roles and other civil developments. What lack is here is a unique kind of motivation as to what specifically make this study unique for the said group. The author needs to go deeper and make very unique demands of the study.
Section 1.4 Economic Violence: Would economic violence be prevalent if for instance, only males were working or even being the top earners in the family? This is because patraichy has also been coined to make breadwinners bosses and that aspect still remain in many communities.
Section 2.1 Study Procedure and Research Ethics: I think this section can just be summarised in a line to say that all ethical clearance expectations were made or followed as per this or that institution. This will give enough space for the author to commence with the engagement without wasting time on these known logistical issues.
Section 2.2.5.Economic Violence Victimization: How does this section differ from 1.4., author needs to justify.
Line 427-429: Is it only about not recognizing their right or does it also delve onto other aspects, like seeking peace at the expense of their own right or is it a cultural blind avoidance. The author needs to avoid stereotyping this since communities are fast becoming aware of their rights according to globalisation movements.
Line 448-449: So avoidance instead of promoting peace it just bring out the undesired perpetration, meaning sitting on a problem or postponing it is not helpful.
Line 475-477: This statement then forces a revision to an earlier argument under spousal economic violence since the former argued that females only were in the receiving end.
Reviewer 5 Report
Comments and Suggestions for Authors
The authors studied a sample of adults in the ultra-Orthodox Jewish community to assess the beliefs that may support gender differences in domestic violence. The paper is interesting and well-written. Of course, there are several limitations, and the authors are partially aware of them. I would suggest critically discussing the reliability of data collected through surveys. Furthermore, what can we learn from this study that can be extrapolated and may also be useful in other cultural contexts? This point may deserve further consideration.
Author Response
Reviewer 4 notes that the paper is interesting and well-written. They suggest critically discussing the reliability of data collected through surveys. Furthermore, the reviewer asks what can be learned from this study that can be extrapolated and may also be useful in other cultural contexts, indicating that this point deserves further consideration.
Response: In consideration the comments provided by all of the reviewers, we have added reference and discussion into the reliability of surveys, particularly among Ultra-Orthodox men who might tend to underreport of victimization to intimate partner violence. Unreliability of data collection through surveys, surrounding sensitive topics in particular, is discussed on the Economic Violence Victimization subsection in the Discussion (p. 13: Line 526). To address this potential bias, we recommend that future research employ qualitative methods to delve into the lived experiences of Orthodox men regarding patriarchy, conflicts in intimate relationships, and violence. This approach would provide a deeper and perhaps more reliable understanding of these issues (p. 14: Line 590).
The manuscript now more effectively addresses cross-cultural issues related to patriarchal beliefs, conflicts, and intimate partner violence in both egalitarian and conservative cultures. This is evident in multiple parts of the manuscript, including the Cultural Context of the Study subsection (p.3, Line 100); the Discussion (pp. 11-12, Line 546); the Conflict Voidance subsection (p. 12: Line 503); and in the Economic Violence Victimization subsection (p. 12: Line 505; p. 13, Line 515).
We have also included a limitation regarding the fact that this study was conducted within a very specific cultural context. We encourage future research to expand the exploration of patriarchal beliefs, as well as intimate partner conflicts and violence, in more liberal and egalitarian cultures (p. 13: Line 543).